# The rs1360780 Variant of *FKBP5*: Genetic Variation, Epigenetic Regulation, and Behavioral Phenotypes

**DOI:** 10.3390/genes16030325

**Published:** 2025-03-11

**Authors:** Marcelo Arancibia, Marcia Manterola, Ulises Ríos, Pablo R. Moya, Javier Moran-Kneer, M. Leonor Bustamante

**Affiliations:** 1Department of Psychiatry, Faculty of Medicine, School of Medicine, Universidad de Valparaíso, Valparaíso 2360002, Chile; marcelo.arancibiame@uv.cl (M.A.);; 2Center for Translational Studies in Stress and Mental Health (C-ESTRES), Faculty of Sciences, Universidad de Valparaíso, Valparaíso 2360102, Chile; mmanterola@uchile.cl (M.M.); pablo.moya@uv.cl (P.R.M.); javier.moran@uv.cl (J.M.-K.); 3Program of Human Genetics, Biomedical Science Institute, Universidad de Chile, Independencia 8380453, Chile; 4Institute of Physiology, Faculty of Sciences, Universidad de Valparaíso, Valparaíso 2360102, Chile; 5Department of Clinical Psychology, School of Psychology, Faculty of Social Sciences, Universidad de Valparaíso, Valparaíso 2341369, Chile

**Keywords:** *FKBP5*, behavioral genetics, environmental stress, genetic variant, genetics, psychiatry

## Abstract

*FKBP5* has been of special scientific interest in the behavioral sciences since it has been involved in the pathophysiology of several mental disorders. It is a gene with pleiotropic effects which encodes the protein FKBP5, a cochaperone that decreases glucocorticoid receptor (GR) affinity for glucocorticoids by competing with FKBP4, altering the GR chaperone complex, and impairing GR activation. As a key modulator of the stress response, FKBP5 plays a critical role in regulating cortisol levels in the organism. The *FKBP5* gene is regulated through a combination of transcriptional, epigenetic, post-transcriptional, and environmental mechanisms, as well as genetic polymorphisms that influence its transcription and stress responsiveness. Notably, the rs1360780 T-allele in *FKBP5* significantly affects FKBP5 regulation and has been linked to stress-related disorders by influencing transcription and stress responsiveness. In this narrative review, we aim to provide an overview of the role played by the single-nucleotide polymorphism rs1360780 in the *FKBP5* locus in gene expression, its epigenetic regulation, and the impact of early stress in its functioning. We discuss some brain regions with differential expression of *FKBP5* and some behavioral phenotypes linked to the locus. The T-allele of rs1360780 is considered a risk variant, as it leads to high *FKBP5* induction, which delays negative feedback and increases GR resistance. This results in states of relative hypercortisolemia and brain morphofunctional alterations, particularly in regions sensitive to glucocorticoid activity during critical periods of neurodevelopment. Additionally, exposure to childhood maltreatment is associated with demethylation of the glucocorticoid response elements of *FKBP5*, further increasing its expression levels. Among the psychological dimensions analyzed in which *FKBP5* is involved are neurocognition, aggression, suicidality, and social cognition. At the level of mental disorders, the gene may play a role in the pathogenesis of post-traumatic stress disorder, depression, and bipolar disorder. In psychotic disorders, its role is less clear. This knowledge enhances the understanding of disease mechanisms that operate through psychopathological dimensions, and highlights the need to design specific, person-centered psychopharmacological and environmental therapeutic interventions.

## 1. Introduction

During the last 15 years, *FKBP5* has been of special scientific interest in the behavioral sciences since it has been involved in the pathophysiology of several mental disorders. It is a gene with pleiotropic effects of great relevance in the functioning of the hypothalamic–pituitary–adrenal (HPA) axis, with an important epigenetic regulation of its expression, especially during childhood, where environmental stress plays a central role.

The *FKBP5* locus is located on the short arm of chromosome 6 (6p21.31) [1] and encodes the protein FKBP5 or FKBP51, a part of the immunophilin family. This designation arose because of its ability to bind the immunosuppressant FK506 [2]. *FKBP5* exhibits various ways of regulating gene expression, such as multiple polyadenylation sites, generating four distinct transcripts by alternative splicing [3]. The encoded protein corresponds to a cochaperone that regulates basic cellular processes such as protein folding and trafficking, cell proliferation and differentiation, and immunoregulation [2]. It modulates glucocorticoid receptor (GR) function and interacts with a number of cytoplasmic proteins, such as the mineralocorticoid receptor (MR) [4], which has an even higher affinity for glucocorticoids than the GR. While MRs are occupied under basal glucocorticoid conditions, GRs respond mostly to glucocorticoids induced by stress responses [4].

Therefore, because of its central and systemic roles and its specific pattern of gene–environment interactions, *FKBP5* is a good model to understand the complex relationship between environmental exposures and genetic variation in the origin of mental disorders. In this narrative review, we aimed to analyze the role played by the rs1360780 variant in the *FKBP5* locus in gene expression, its epigenetic regulation, and the impact of early stress in its functioning. We discuss some brain regions with differential expression of *FKBP5* and some psychological and psychiatric phenotypes linked to the locus. Finally, we describe some evidence of intergenerational stress inheritance via epigenetic modifications of *FKBP5*.

## 2. *FKBP5* Plays a Fundamental Role in the Molecular Stress Response

In stressful events, the brain cortex and the limbic system activate the primary effector of the stress response, the HPA axis. Briefly, the paraventricular nucleus of the hypothalamus secretes adrenocorticotrophin-releasing hormone (CRH) and arginine-vasopressin, which in turn triggers the secretion of adrenocorticotrophin (ACTH) from the adenohypophysis, activating the production of cortisol and catecholamines from the adrenal glands, whose function is the recovery of homeostasis after stress events [2]. Likewise, cortisol exerts negative feedback on the hypothalamus and adenohypophysis to regulate their secretion [5]. CRH also exerts its action on the locus coeruleus, activating the sympathetic response to stress [6]. In the brain, GR activation leads to rapid inhibition of the expression of genes encoding CRH and ACTH [2].

The protein FKBP5 is located in both the cytoplasm and cell nucleus as part of the GR complex, a ligand-dependent transcription factor [7]. This receptor is regulated by a multiprotein complex comprising various folding chaperone proteins, including heat shock proteins (Hsps) and FK506 binding proteins [8,9,10]. FKBP5 presents a tetratricopeptide domain, by which it binds to cochaperone proteins, such as P23 and Hsp90, which are associated with the GR. FKBP5 regulates GR signaling by two main pathways. First, its binding to the GR induces a conformational change that decreases its affinity for cortisol [10,11]. Second, FKBP5 promotes the cytoplasmic translocation of an inactive form of the GR (the beta-isoform), a step that precedes its activation and nuclear import [12]. Upon binding to cortisol, FKBP5 is replaced by FKBP4 (also known as FKBP52), which promotes the recruitment of dynein proteins to the complex and the translocation of the GR to the nucleus [11].

In the nucleus, the GR binds to sequences known as glucocorticoid response elements (GREs) [10], promoting *FKBP5* expression [13]. This regulation also involves activity on transcription factors that are independent of GREs. In addition to glucocorticoids, other steroid hormones can upregulate *FKBP5* expression. GREs are located in upstream regions and within the *FKBP5* gene, including in the enhancer regions of introns 2, 5, and 7 [14]. Epigenetic marks in *FKBP5*, such as methylation of histone H3 lysine 4, acetylation of H3 lysine 27, and trimethylation of H3 lysine 36, indicate an open three-dimensional chromatin conformation even before steroid binding occurs [15]. That is, non-covalent modifications in histones will ultimately determine changes in the three-dimensional structure of chromatin that will favor or limit the access of transcription factors to DNA and thus modify gene expression.

The inhibitory function that FKBP5 exerts on GR activity establishes an ultra-short negative feedback loop, where increased *FKBP5* expression diminishes glucocorticoid signaling [16,17]. Notably, *FKBP5* expression is variable depending on the tissue and developmental stage [10].

For its part, upon glucocorticoid binding to the MR under baseline conditions, FKBP5 is replaced by FKBP4, which promotes the translocation of the MR-Hsp90 complex into the nucleus and subsequent DNA binding [4].

## 3. Genetic Polymorphisms That Influence *FKBP5* Transcription and Stress Responsiveness: Role of the rs1360780 T-Allele

In individuals of European as well as of African descent, a functional haplotype of FKBP5 containing 18 single-nucleotide polymorphisms (SNPs) has been observed. Among them, rs1360780, a biallelic SNP located in intron 2, 50 kb downstream of the transcription start site [18,19], is the variant most strongly associated with increased mRNA expression following GR action. This effect may be attributed to alterations in the GREs that enhance activation [10].

rs1360780 has two alleles, C and T. The minor allele (T) has a frequency of about 20% [20]. It is the most studied variant in humans in relation to alterations in the stress response, and it is located in an enhancer region close to a GRE. The presence of the T-allele is associated with structural DNA and chromatin modifications that play a crucial role in *FKBP5* transcription. Sequences containing this variant have a higher affinity for the TATA-box binding protein, which facilitates the direct contact of intron 2 with the transcription start site [8,14,21,22]. This interaction is mediated by specific three-dimensional chromatin configurations that promote gene expression [15,21,22].

Binder et al. [8] evaluated the effect of rs1360780 variants in 294 subjects with mood disorders and 339 healthy controls, all of Caucasian ethnicity, with more than 90% of German origin. They found that protein FKBP5 levels were 2-fold higher in carriers of the TT genotype of rs1360780. Likewise, in healthy controls, *FKBP5* mRNA levels in peripheral blood cells were higher in carriers of the TT genotype. These results are consistent with those subsequently found by Klengel et al. [21] in a cohort of African American participants. In their research, of particular relevance in the field, the authors performed functional tests that corroborated the effect of the T-allele on transcription and chromatin structure, as well as its epigenetic regulation.

This regulation is complex and depends on the activity of specific sequence enhancers and epigenetic changes, including DNA methylation patterns and covalent modifications of histone proteins. Additionally, the activity of the *FKBP5* transcriptional machinery is modulated by DNA-binding proteins like CTCFs and cohesins [14]. In healthy human controls, the T-allele of rs1360780, also known as a high-induction allele, has been shown to be associated with a GR resistance mechanism [23] explained by increased gene expression. This GR resistance triggers a prolonged cortisol response leading to relative hypercortisolemia [3,16]. The hypercortisolism can be explained by impaired feedback inhibition, which should be most pronounced when the HPA axis is activated during stress-related events [24].

For its part, the C-allele is characterized by less physical contact of the intron 2 GRE with the FKBP5 transcriptional start site, so it would not alter the negative feedback associated with GR activity once the stress response begins [24]. There is emerging evidence that under non-stress conditions, the expression of *FKBP5* is regulated by the MR and not by the GR [4]. However, the direct relationship between the rs1360780 T-allele and the functioning of the MR has not yet been studied.

## 4. Effect of Environment on Epigenetic Regulation of *FKBP5*: Case of Early Stress

Epigenetic modifications regulate gene expression and are essential for the development of organisms. They can be induced by environmental phenomena, such as feeding, exposure to substances, and psychological stress. Thus, the connection between the environment and the genome is mediated by epigenetic mechanisms. The three major epigenetic mechanisms are post-translational modifications of histone proteins (i.e., methylation, acetylation, ubiquitination) that promote structural changes of chromatin to allow greater or lesser access of transcription factors to DNA; regulation by non-coding RNAs, responsible for modulating translation and mRNA degradation; and DNA methylation and hydroxymethylation. The last mechanism occurs in CpG islands, regions rich in cytosine-guanine dinucleotides [25]. Methylated sequences are silenced, as methylation prevents the binding of transcription factors essential for gene expression; however, this is not always associated with a lower level of gene expression, especially in the methylation of intronic regulatory regions [26]. The effect of methylation will depend on the function associated with the methylated sequence, for instance, if they involve an enhancer or silencer. Furthermore, the methylation phenomenon is allele-specific, i.e., it also depends on the genetic sequence and can be affected by single-nucleotide variants. A total of 40% of the methylated CpG sites in the mammalian genome are found in intragenic regions, and 34% of all intragenic CpG islands are methylated in the human brain [7].

In the case of rs1360780, it has been observed that T-allele carriers exposed to childhood maltreatment exhibit significant demethylation in the GRE of intron 2 of *FKBP5* in peripheral lymphocytes. These two factors reinforce the contact between the GRE and the transcription start site, promoting *FKBP5* expression; that is, they have a net effect of disinhibition of *FKBP5* transcription induced by GR signaling, which has been associated with resistance or lower sensitivity of the GR [21,22]. This GRE is part of an enhancer sequence that, when demethylated, promotes *FKBP5* expression [3]. Demethylation is mediated by GR activity, as it activates base excision repair mechanisms that replace methylated sites with non-methylated sites [3]. This mechanism has not been replicated in C- allele carriers [10,21]. Similar results have been found in human hippocampal cell lines exposed to glucocorticoids during proliferation and differentiation [22]. In turn, intron 7 demethylation is higher in T-allele carriers [21], which reinforces the idea of allele-specific changes in methylation patterns [27,28,29]. This phenomenon has been termed the “double hit theory of *FKBP5* disinhibition” [3,27,30], as it suggests a state of *FKBP5* disinhibition stemming from the risk genetic variant (T-allele) and early stress-related epigenetic modifications (Figure 1). Interestingly, the possible epigenetic effect of environmental stress on *FKBP5* methylation may only occur in models of childhood maltreatment, as it has not been corroborated in adult stress. This raises the interesting possibility that sensitive periods of development are more affected by epigenetic influences.

## 5. *FKBP5* Is Differentially Expressed in Different Brain Regions and Is Associated with Morphofunctional Changes

*FKBP5* induction by the GR is conserved across species, but expression patterns are highly variable between species and between tissues [2]. Although the HPA axis articulates the central response to stress, there are afferents coming from structures such as the prefrontal cortex, which promotes the inhibition exerted by the hippocampus on the hypothalamus during environmental stress, or the amygdala, a structure that, in the opposite direction, stimulates the hypothalamic response to stress. All these regions are sensitive to stress and, not surprisingly, all have high levels of GR [31] and *FKBP5* [32] expression. Early or chronic adverse experiences can lead to HPA axis modifications that are associated with alterations such as a reduction in the volume of the hippocampus [6], a structure particularly sensitive to early stress and the effect of cortisol. In turn, the hippocampus regulates the HPA axis by inhibiting the hypothalamus [6] and consequently decreasing cortisol levels; conversely, the amygdala exerts a mainly excitatory action on the HPA axis [6,33]. In murine models, it has been shown that corticosterone exposure-induced GR activity is associated with *FKBP5* demethylation in the hippocampus and hypothalamus, but also in peripheral blood lymphocytes [34], which suggests that this tissue is suitable for the study of epigenetic changes at the level of this sequence in the brain [10].

In healthy participants, Hirakawa et al. [35] studied the potential morphological and psychological effects of six SNPs using voxel-based morphometry. rs1360780 T-allele carriers had a larger gray matter volume in the right amygdala. The authors hypothesize that these changes associated with the presence of *FKBP5* variants lead to the observed anxious and depressive symptomatology. In the same line, Wesarg et al. [36] studied the effects of the interaction between rs1360780 and child abuse on the resting-state functional connectivity between the amygdala and other regions of the salience network (774 European adults from the general population); they verified a significant interaction effect for functional connectivity between the right centromedial amygdala and right posterior insula, which was stronger in people with the TT genotype who suffered from abuse. These results emphasize the relevance of structures in charge of interoceptive awareness processing such as the amygdala and insula, which may show increased coordination in TT genotype carriers. Although findings on amygdala lateralization are scarce, one systematic review concluded that the right amygdala is more involved in global processing of emotional stimuli, i.e., holistic aspects of emotional stimuli [37]. Another neuroimaging study carried out by Fujii et al. in a non-clinical population found that T-allele carriers showed significantly higher mean diffusivity values in the dorsal anterior cingulate cortex and posterior cingulate cortex, suggesting altered white matter integrity in these regions. According to the authors, these results might be interpreted as loss of neurons, axons, and dendrites [38]. More specifically, regarding cellular differences in humanized mouse lines, Nold et al. found that, in T-allele carriers, the transcriptional response of *FKBP5* to glucocorticoids was highest in astrocytes, followed by microglia and neurons. These results highlight the potential relevance of astrocytes in the stress response [39].

In people with depression, Han et al. [40] found that *FKBP5* intron 7 demethylation was associated with both the presence of the T-allele and a reduction in prefrontal gray matter volume. These findings are consistent with those verified by Tozzi et al. [41], who reported that, in people with depression, the interaction between the T-allele of rs1360780 and the presence of childhood maltreatment accounted for reduced activity in the insula, parahippocampal gyrus, posterior cingulate cortex, and inferior frontal gyrus; these results were not found in CC homozygotes. A study comparing African American women with and without post-traumatic stress disorder (PTSD) concluded that carriers of the TT genotype of rs1360780 with the pathology demonstrated poorer cingulum connectivity compared to the other genotype and diagnostic groups [42].

## 6. Behavioral Phenotypes Associated with Variations in the *FKBP5* Locus

Stress is a nonspecific risk factor for the development of mental disorders. Not all individuals under stress develop mental disorders, suggesting that genetic makeup is a key determinant of outcome. *FKBP5* plays a central role in stress physiology, and some risk variants may confer increased sensitivity to stressful events, which could decrease the fitness of carriers. However, from an evolutionary perspective, Matosin et al. [3] hypothesize that the persistence of the *FKBP5* risk haplotype in the population is associated with increased anxiety consequent to stressful life events promoting survival behaviors; jointly, some clinical phenotypes associated with *FKBP5* variants may not have reduced fertility, since the psychopathology associated with risk alleles occurs in adverse environmental conditions but not in favorable ones; i.e., from the gene–environment interaction approach, it would correspond to a model of differential sensitivity [43].

The T-allele is associated with high induction of *FKBP5*. In most studies, it is implicated in different psychopathological dimensions and psychiatric phenotypes [10]. However, it has also been associated with a better response to antidepressants in depressive disorders [8,44]. Therefore, the outcome will largely depend on the environmental circumstances interacting with the genetic makeup, including life span, time of exposure, and the nature of the environmental factor.

Due to the high expression of *FKBP5* in structures such as the hippocampus, amygdala, and prefrontal cortex, the variety of clinical phenotypes associated with locus variants is wide, and they are related to alterations in memory, learning, fear extinction, and emotional regulation in general [45]. There is post-mortem evidence of increased *FKBP5* expression in cortical and hippocampal regions of people with mood and psychotic disorders. Compared to healthy controls, increased *FKBP5* mRNA expression has been identified in the prefrontal cortex of people with schizophrenia and bipolar disorder [46]. Ferrer et al. [47] note that negative feedback to the HPA axis is regulated by *FKBP5* polymorphisms independently of psychiatric diagnosis. Thus, the role of *FKBP5* has been implicated in several stress-related mental disorders more directly, such as PTSD [48] and major depressive disorder [49,50], and more indirectly, such as bipolar disorder [51] and schizophrenia [52]. Given the diversity of clinical phenotypes associated with *FKBP5* function, pleiotropic effects could be assumed [10]. *FKBP5* is a central determinant of HPA axis; therefore, its functioning is of major relevance in stress-related disorders such as acute stress disorder and PTSD. Likewise, a number of other genes, such as *CRHR1* and *NR3C1*, which encode for the CRHR and GR, respectively, are essential as well. In PTSD, there is an increase in CRH secretion associated with an increase in negative feedback to the HPA axis [53]. This phenomenon will also be determined by the presence of genetic risk variants and their epigenetic changes for the development of HPA axis dysfunction [54].

Some studies have found differences in the stress response according to sex, an effect that is mediated by polymorphisms in *FKBP5*. An interaction between rs1360780, sex, and early stress has been found in mice [55] and in humans [56,57], which is greater in female carriers of the T-allele. In fact, *FKBP5* has been suggested as a female-specific biomarker for prolonged cortisol load [55]. This evidence sheds light on the putative mechanisms of sexual dimorphism in stress-related phenotypes.

### 6.1. Psychopathological Dimensions

In healthy participants, *FKBP5* variants have been implicated in dimensions such as aggression [58], social cognition [17,36], and neurocognition [17,59]. In a sample of healthy participants, Ferrer et al. [17] verified that childhood physical abuse, childhood physical neglect, and the T-allele of rs1360780 predicted performance on social cognition tasks. Carriers of the T-allele who had experienced physical abuse performed better on social cognition than individuals with the CC genotype. Although this result is contrary to what is usually reported, the authors point out that the T-allele, in this case protective, could moderate a resilience mechanism in social cognition in people exposed to childhood maltreatment.

Binder et al. [23] developed a cross-sectional study (n = 762) to analyze the association between different *FKBP5* variants and the presence of post-traumatic symptoms in adults without psychiatric disorders who had experienced childhood maltreatment. Four SNPs (rs3800373, rs9296158, rs1360780, and rs9470080) were found to have a significant interaction with the severity of abuse in predicting levels of post-traumatic symptomatology, with the highest magnitude found for rs9296158 (intron 5). This association was maintained after controlling for variables such as age, sex, depressive symptomatology, exposure to trauma not linked to childhood maltreatment, and ancestry. The SNPs analyzed present a very high level of linkage disequilibrium and all of them are associated with increased *FKBP5* mRNA levels in the presence of cortisol [8]. These results show the possibility that there is an interaction between different SNPs of the gene and that this explains the observed phenotypes. In this sense, the AGCT haplotype for the previous four SNPs confers a risk of PTSD [60], whereas the CATT haplotype (rs1360780-rs9296158-rs4713916-rs9470080) confers a risk of major depressive disorder [61].

### 6.2. Mental Disorders

The rs1360780 SNP has the best-characterized mechanism in relation to *FKBP5* polymorphisms and their role in glucocorticoid signaling [7], while the T-allele is the most studied variant in psychiatric phenotypes. This has been most consistently corroborated in conditions such as major depressive disorder [18]. In PTSD, on the other hand, the C-allele has been found to be linked to an increased risk of presenting with the disorder as well [7,62]. In healthy [23] and depressed adults [63], T-allele carriers have a significantly decreased suppression of the HPA axis when undergoing the dexamethasone suppression test. This effect could be the opposite in T-allele carriers suffering from PTSD, in whom increased GR sensitivity is also observed [2,63], in contrast to phenotypes associated with mood disorders, where GR resistance has been reported. The mechanisms of this difference are not entirely clear. In a study of 2157 adults from the general population in Germany, Appel et al. [50] found a significant interaction between the TT genotype of rs1360780 and the presence of sexual, emotional, and physical child abuse to explain the risk of depression. Carriers of this genotype had a significantly higher risk of depression than those with the CC/CT genotypes.

In bipolar disorder, the study of *FKBP5* is more incipient. In a systematic review of genome-wide gene expression studies from post-mortem human brain tissue samples of people with bipolar disorder, 382 differentially expressed genes were identified, but only 11, including *FKBP5*, exceeded the significance level after multiple testing correction in the prefrontal cortex. This supports the hypothesis of the relevance of *FKBP5* in the pathology [64]. As has been shown in unipolar depression, carriers of the T-allele with bipolar disorder present an alteration in the stress response that, at the clinical level, has been linked to the presence of depressive episodes and suicidality [65]. In a sample of patients with unipolar and bipolar depression, it was concluded that in TT homozygotes, both GR signaling and HPA axis reactivity were more controlled by the interaction between *FKBP5* and the GR than in carriers of the CT or CC genotypes [8]. Regarding the effects of childhood maltreatment on *FKBP5* methylation, in people with bipolar disorder and major depressive disorder, the claims of Klengel et al. [21,30] have been verified. From a sample of 61 patients with bipolar disorder, Saito et al. [51] observed that carriers of the T-allele of rs1360780 who suffered childhood neglect or emotional abuse expressed lower levels of *FKBP5* methylation. Accordingly, in a study with 3965 participants, intron 7 demethylation was observed in depressed patients carrying the T-allele of rs1360780. However, demethylation was not significantly associated with childhood maltreatment or a depressive state [66].

A high proportion of patients with psychotic disorders present alterations in the HPA axis. It has been postulated that glucocorticoid secretion is associated with an increase in dopaminergic activity in specific brain regions, which could cause psychotic symptoms [67]. However, research on *FKBP5* is scarce and less conclusive in this population [52]. A study in 808 healthy adults could not corroborate an association between *FKBP5* risk haplotypes (CAT for rs3800373-rs9296158-rs1360780, and TA for rs9470080-rs4713916) and schizotypy or psychotic-like experiences [68]. For its part, an investigation with a Belgian sample of healthy children of people with psychotic disorders and patients with psychotic disorders demonstrated that the *FKBP5* polymorphisms that had a significant interaction with child abuse on psychosis were the A-alleles of rs4713916 and rs9296158 [67]. A research that included 48 patients with psychotic disorders, 50 unaffected siblings, and 46 healthy controls showed that there was significant demethylation determined by the T-allele of rs1360780 in the patient group, while there was an interaction between the T-allele and child abuse that had a significant effect on *FKBP5* demethylation, an effect that was only verified in the control group. Finally, a study conducted in England including 291 first-episode psychotic cases and 218 healthy controls demonstrated that the association between rs1360780 and the presence of psychosis was only found when the analysis was adjusted for cannabis use and parental separation [69].

## 7. Intergenerational Stress Inheritance via Epigenetic Modifications in *FKBP5*

Parental stress and trauma associated with chronic stress lead to epigenetic changes in the *FKBP5* gene that can be passed on to subsequent generations. Indeed, inherited stress-induced DNA methylation changes in *FKBP5* are associated with increased cortisol levels and a deregulated stress response in the offspring. A remarkable study of Holocaust survivors (n = 32) and their offspring (n = 22) found that parental trauma influenced *FKBP5* intron 7 methylation patterns in the next generation (F1). Survivors showed increased methylation at the third GRE of intron 7, while their children had decreased methylation compared to controls, possibly reflecting an adaptation to altered glucocorticoid levels [70]. Additionally, offspring with the *FKBP5* rs1360780 A-allele showed a negative correlation between childhood abuse and methylation at the first GRE of intron 7, whereas GG genotype carriers exhibited a positive correlation, suggesting epigenetic responses to personal or inherited trauma [70]. These contrasting epigenetic modifications in *FKBP5* observed in both parents and offspring may facilitate a biological adaptation in the children, leading to altered stress responses compared to their parents.

Research has demonstrated that paternal stress can induce epigenetic changes in germ cells, particularly in regulatory regions of key HPA axis genes such as *NR3C1* [71] and *CRHR1* [72]. Whether chronic stress induces similar epigenetic changes in *FKBP5* within sperm and/or oocytes remains a critical area for further investigation. Understanding how methylation changes in this gene are inherited by offspring is essential for elucidating the mechanisms underlying intergenerational stress transmission. Regardless, the role of *FKBP5* in perpetuating intergenerational cycles of stress vulnerability is undoubtable.

## 8. Conclusions

*FKBP5* plays a central role in the stress response by increasing its expression in response to GR activation. The T-allele of the rs1360780 SNP is considered a risk variant because it determines a high induction of *FKBP5*, delaying negative feedback and increasing GR resistance, which leads to states of relative hypercortisolemia and brain morphofunctional changes, especially in regions sensitive to glucocorticoid activity and during specific periods of neurodevelopment. This is expected to cause an altered response of the HPA axis to stress and, ultimately, behavioral phenotypes. However, experiences of early adversity add complexity to this hypothesis. Exposure to childhood maltreatment is associated with demethylation of the GREs of *FKBP5*, increasing its expression level. Among the psychological dimensions analyzed in which *FKBP5* is involved are neurocognition, aggression, suicidality, and social cognition. At the level of mental disorders, the gene plays a role in the pathogenesis of PTSD, depression, and bipolar disorder. In psychotic disorders, its role is less clear.

Therefore, current evidence supports that *FKBP5* plays a cross-cutting role in multiple mental disorders. Although the directionality of the effect of *FKBP5* variants in different pathologies is not entirely consistent, this fact recalls the relevance of the gene–environment interaction in determining the effect of a specific genetic variant. In this line, the expression of *FKBP5* has an important epigenetic regulation, and this has been well characterized. This knowledge can contribute to the understanding of disease mechanisms that operate dimensionally, as well as to the design of specific, person-centered psychopharmacological and environmental therapeutic interventions.

## Figures and Tables

**Figure 1 genes-16-00325-f001:**
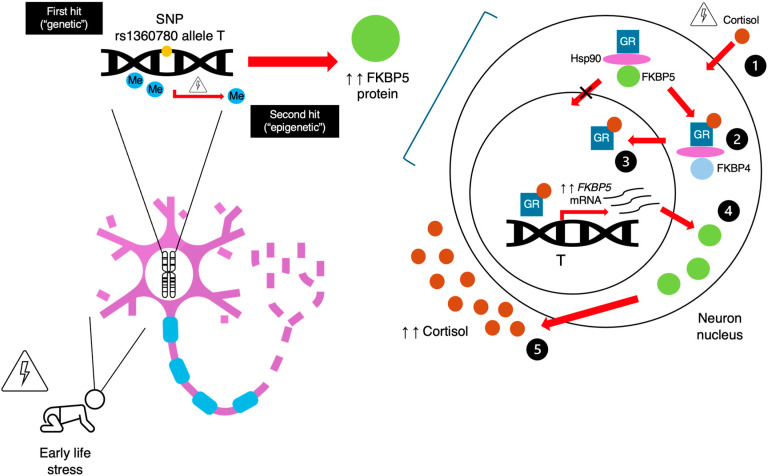
Schematic view of the “double hit theory of *FKBP5* disinhibition”. SNP: single-nucleotide polymorphism; FKBP5: FKBP5 protein; FKBP4: FKBP4 protein; Hsp90: heat shock protein 90; GR: glucocorticoid receptor.

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
