# Peer review of "The rs1360780 Variant of FKBP5: Genetic Variation, Epigenetic Regulation, and Behavioral Phenotypes"

_genes, 2025, doi:10.3390/genes16030325_

Round 1
Reviewer 1 Report
Comments and Suggestions for Authors
Arancibia et al. presented the role of the rs1360780 variant in the FKBP5 locus in regulating gene expression, its epigenetic modulation, and the impact of early stress on its function. This is an insightful review article that discusses brain regions with differential FKBP5 expression, psychological and psychiatric phenotypes associated with this locus, and evidence supporting the intergenerational inheritance of stress via epigenetic modifications of FKBP5. However, there are some aspects that need to be addressed to further enhance the manuscript.
1. The issue of laterality discussed in Section 5 is very interesting; however, the reviewer believes the discussion would be even more compelling if it delved into the neural circuits involved.
2. Regarding morphological changes, does the volume change occur in neurons only, or does it also include glial cells? Additionally, do the number and morphology of synapses and spines change as well?
3. MR is only briefly mentioned in the introduction, but it is necessary to discuss the relationship between MR and the rs1360780 variant of FKBP in more detail, especially in comparison to GR, within the text.
4. Is the inheritance to offspring limited to the F1 generation only?
5. A discussion of the effect of sex differences on the rs1360780 variant in FKBP5 would be intriguing.

Author Response
Comment (C): Arancibia et al. presented the role of the rs1360780 variant in the FKBP5 locus in regulating gene expression, its epigenetic modulation, and the impact of early stress on its function. This is an insightful review article that discusses brain regions with differential FKBP5 expression, psychological and psychiatric phenotypes associated with this locus, and evidence supporting the intergenerational inheritance of stress via epigenetic modifications of FKBP5. However, there are some aspects that need to be addressed to further enhance the manuscript
Response (R): We appreciate the work done by the reviewer, which highlights fundamental aspects of the discussion.
C: The issue of laterality discussed in Section 5 is very interesting; however, the reviewer believes the discussion would be even more compelling if it delved into the neural circuits involved.
R: We appreciate the reviewer's comment. We add to the discussion the results of Wesarg et al., Schaan et al., and Baas et al. The former on functional aspects of the right amygdala and its interaction with the SNP rs1360780 and the latter on the implications of amygdala lateralization.
C: Regarding morphological changes, does the volume change occur in neurons only, or does it also include glial cells? Additionally, do the number and morphology of synapses and spines change as well?
R: We appreciate the reviewer's comment. To our knowledge, we have no evidence in humans demonstrating such a particular effect on synapsis structure. However, we have incorporated the results of the study by Fujii et al.: “Another neuroimaging study carried out by Fujii et al. in non-clinical population [38] found that rs1360780 T carriers showed significantly higher mean diffusivity values in the dorsal anterior cingulate cortex and posterior cingulate cortex, suggesting altered white matter integrity in these regions. According to the authors, these results might be interpretated as loss of neurons, axons, and dendrites.” Additionally, we have incorporated the conclusions of Nold et al.: “More specifically, regarding cellular differences in humanized mouse lines, Nold et al. found that, in T allele carriers, the transcriptional response of FKBP5 to glucocorticoids was highest in astrocytes, followed by microglia and neurons. These results highlight the potential relevance of astrocytes in the stress response [39].”
C: MR is only briefly mentioned in the introduction, but it is necessary to discuss the relationship between MR and the rs1360780 variant of FKBP in more detail, especially in comparison to GR, within the text.
R: We appreciate the reviewer's comment. To our knowledge, there are no studies directly linking the T-allele of rs1360780 to MR functioning. We have incorporated an explanation of MR nuclear translocation in the Introduction. Additionally, we have pointed out: “There is emerging evidence that under non-stress conditions, the expression of FKBP5 is regulated by MR and not by GR [4]. However, the direct relationship between the rs1360780 T-allele of the functioning of MR has not yet been studied.”
C: Is the inheritance to offspring limited to the F1 generation only?
R: We appreciate the reviewer's comment. In fact, the study of Yehuda et al. included F0 and F1. We have incorporated the sample sizes and the concept “F1”.
C: A discussion of the effect of sex differences on the rs1360780 variant in FKBP5 would be intriguing.
R: We appreciate the reviewer's comment. We have included in the discussion three references in this regard (Nold et al. 2022; Dackis et al. 2012; VanZomeren-Dohm et al. 2015) and their main conclusions.

Reviewer 2 Report
Comments and Suggestions for Authors
- Please provide more details on the studies that investigate the single-nucleotide polymorphisms that are associated with the transcription of FKBP5. Also describe what are the populations included in each study, such as their race/ethnicity.
- In line 119, please cite more papers.
- In line 112, please provide more evidence on how rs1360780 is the SNP that most strongly associate with the mRNA expression.

Author Response
Comment (C): Please provide more details on the studies that investigate the single-nucleotide polymorphisms that are associated with the transcription of FKBP5. Also describe what are the populations included in each study, such as their race/ethnicity.
Response (R): We appreciate the reviewer's comment. We have included more details on this issue: “Binder et al. [8] evaluated the effect of rs1360780 variants in 294 subjects with mood disorders and 339 healthy controls, all of Caucasian ethnicity and more than 90% of German origin. They found that FKBP5 protein levels were 2-fold higher in carriers of the TT genotype of rs1360780. Likewise, in healthy controls, FKBP5 mRNA levels in peripheral blood cells were higher in carriers of the TT genotype. These results are consistent with those subsequently found by Klengel et al. [21] from a cohort of African-American participants. In the latter research, of particular relevance in the field, the authors performed functional tests that corroborated the effect of the T-allele on transcription and chromatin structure, as well as its epigenetic regulation.”
C: In line 119, please cite more papers.
R: We appreciate the reviewer's comment. In addition to the references of Klengel et al. 2013A and Klengel et al. 2013B, we have included Binder et al. 2004 and Pakinaaho et al. 2010.
C:In line 112, please provide more evidence on how rs1360780 is the SNP that most strongly associate with the mRNA expression.
R: We appreciate the reviewer's comment. In the second paragraph of point 3, we described the potential structural mechanism that explains the increased expression levels of FKBP5: “The presence of the T- allele is associated with structural DNA and chromatin modifications that play a crucial role in FKBP5 transcription. Sequences containing this variant have a higher affinity for the TATA-box binding protein, which facilitates direct contact of intron 2 with the transcription start site [21,22]. This interaction is mediated by specific three-dimensional chromatin configurations that promote gene expression [15,21,22].” We have incorporated further description of the studies obtaining these conclusions in the additional paragraph that was added as a reviewer's suggestion (third paragraph of point 3).

Reviewer 3 Report
Comments and Suggestions for Authors
The review presents FKBP5 and its role in the stress response and psychiatric disorders. It provides an overview of the gene’s functions, its involvement in gene-environment interactions, and how they contribute to stress regulation and mental health. The focus on FKBP5’s epigenetic regulation and its link to psychiatric phenotypes is of medical interest.
However, there are a few areas that could be improved, as detailed in the following suggestions and comments.
Abstract Section
- while the abstract is too dense, the aim of the study is not explicitly stated in the abstract.
- the abstract does not specify the type of review. Since there’s no methodology described, it aligns with a narrative review. The authors should include this clarification.
- Introduction Section
- the authors should clarify both the objective and type of the review to ensure the reader understands what to expect.
- FKBP5 plays a fundamental role in the molecular stress response
- the title suggests a focus on the rs1360780 variant and behavioral phenotypes, but this is missing from the section.
- the explanation about FKBP5 being replaced by FKBP4 could be expanded for clarity, particularly how this switch facilitates GR nuclear translocation and why this is important for gene regulation.
- it is missing a brief explanation of the broader implications of FKBP5 dysregulation in stress-related disorders.
- the section mentions epigenetic marks but does not fully explain how they regulate FKBP5 expression.
- Genetic polymorphisms that influence FKBP5 transcription and stress responsiveness: role of the rs1360780 T-allele
- the authors should explain how FKBP5 overexpression impairs GR sensitivity and how this alters the HPA axis response.
- how does the T allele specifically modify the methylation landscape and histone acetylation?
- minor editing error: line 124 ”the activity of the FKBP5 transcriptional machinery is modeulated by DNA binding proteins” changes to ……..modulated…………
- The effect of environment on epigenetic regulation of FKBP5: the case of the early stress
- the author should explain why the T allele is particularly susceptible to demethylation in response to early stress.
- providing a brief comparison with other genes involved in the stress response (for example, NR3C1 or CRHR1) could give more context.
- a schematic representation of the "double hit theory" would be very helpful for visualisation.

The manuscript requires major revisions before it can be considered for publication.
Author Response
Comment ( C):The review presents FKBP5 and its role in the stress response and psychiatric disorders. It provides an overview of the gene’s functions, its involvement in gene-environment interactions, and how they contribute to stress regulation and mental health. The focus on FKBP5’s epigenetic regulation and its link to psychiatric phenotypes is of medical interest.
Response (R ): . We appreciate the general commentary on our article and the thorough work done by the reviewer.
C: However, there are a few areas that could be improved, as detailed in the following suggestions and comments.
Abstract Section
- while the abstract is too dense, the aim of the study is not explicitly stated in the abstract.
R: We appreciate the reviewer's comment. We have explicitly incorporated the objective of the article: “In this narrative review, we aimed to provide an overview of the role played by the single-nucleotide polymorphism rs1360780 in the FKBP5 locus in the gene expression, its epigenetic regulation, and the impact of early stress in its functioning. We discuss some brain regions with differential expression of FKBP5 and some behavioral phenotypes linked to the locus.”
C: the abstract does not specify the type of review. Since there’s no methodology described, it aligns with a narrative review. The authors should include this clarification.
R: We appreciate the reviewer's comment. We have explicitly incorporated the type of review: “In this narrative review…”
C: Introduction Section
- the authors should clarify both the objective and type of the review to ensure the reader understands what to expect.
R: We appreciate the reviewer's comment. We have explicitly incorporated the aim of the article and the type of review: “In this narrative review…”
C: FKBP5 plays a fundamental role in the molecular stress response
- the title suggests a focus on the rs1360780 variant and behavioral phenotypes, but this is missing from the section.
R: We appreciate the reviewer's comment. We have focused on rs1360780 SNP, particularly in T variant in the third section.
C: the explanation about FKBP5 being replaced by FKBP4 could be expanded for clarity, particularly how this switch facilitates GR nuclear translocation and why this is important for gene regulation.
R: We appreciate the reviewer's comment, since this mechanism is of utmost importance to understand the biological relevance of FKBP5. Regarding the interaction between FKBP5/FKBP4, in the second section we included: “FKBP5 presents a tetratricopeptide domain, by which it binds to cochaperone proteins, such as P23 and Hsp90, which are associated with the GR. FKBP5 regulates GR signaling by two main pathways. First, its binding to the GR induces a conformational change that decreases its affinity for cortisol [10,11]. Second, FKBP5 promotes the cytoplasmic translocation of an inactive form of GR (the beta-isoform), a step that precedes its activation and nuclear import [12]. Upon binding to cortisol, FKBP5 is replaced by FKBP4 (also known as FKBP52), which promotes the recruitment of dynein proteins to the complex and the translocation of GR to the nucleus [11].” In addition, we have incorporated an explanation of mineralocorticoid receptor nuclear translocation.
C: it is missing a brief explanation of the broader implications of FKBP5 dysregulation in stress-related disorders.
R: We appreciate the reviewer's comment. We have included: “FKBP5 is a central determinant of HPA axis functioning, therefore, its functioning is of major relevance in stress-related disorders such as acute stress disorder and PTSD. Likewise, a number of other genes, such as CRHR1 and NR3C1, which encode for CRHR receptor and GR, respectively, are essential as well. In PTSD there is an increase in CRH secretion associated with an increase in negative feedback to the HPA axis [53]. This phenomenon will also be determined by the presence of genetic risk variants and their epigenetic changes for the development of HPA axis dysfunction [54].” We have included the follow references: Daskalakis et al. 2013 and Alahmad et al. 2025.
C: the section mentions epigenetic marks but does not fully explain how they regulate FKBP5 expression.
R: We appreciate the reviewer's comment. For clarity, we have included: “That is, non-covalent modifications in histones will ultimately determine changes in the three-dimensional structure of chromatin that will favor or limit the access of transcription factors to DNA and thus modify gene expression.” In addition, in the first paragraph of the fourth section we explained how CpG methylation modulates the gene expression.
C: Genetic polymorphisms that influence FKBP5 transcription and stress responsiveness: role of the rs1360780 T-allele
- the authors should explain how FKBP5 overexpression impairs GR sensitivity and how this alters the HPA axis response.
R: We appreciate the reviewer's comment. We have included: “The hypercortisolism can be explained by an impaired feedback inhibition, which should be most pronounced when the HPA axis is activated during stress-related events [24].”
C: how does the T allele specifically modify the methylation landscape and histone acetylation?
R: We appreciate the reviewer's comment. This comment points to a gap in current knowledge, as studies on the subject are very scarce. In the fourth section we included: “Demethylation is mediated by GR activity (which in turn is largely determined by the FKBP5 protein), as it activates base excision repair mechanisms that replace methylated sites with non-methylated sites [3]. This mechanism has not been replicated in C- allele carriers [10,21]. Similar results have been found in human hippocampal cell lines exposed to glucocorticoids during proliferation and differentiation [22]. In turn, intron 7 demethylation is higher in T-allele carriers [21], which reinforces the idea of allele-specific changes in methylation patterns [27–29].”
C: minor editing error: line 124 ”the activity of the FKBP5 transcriptional machinery is modeulated by DNA binding proteins” changes to ……..modulated…………
R: We appreciate the reviewer's comment. We have corrected this error.
C: The effect of environment on epigenetic regulation of FKBP5: the case of the early stress
- the author should explain why the T allele is particularly susceptible to demethylation in response to early stress.
R: We appreciate the reviewer's comment. This comment points to a gap in current knowledge, as studies on the subject are very scarce. In this section we included: “Demethylation is mediated by GR activity (which in turn is largely determined by the FKBP5 protein), as it activates base excision repair mechanisms that replace methylated sites with non-methylated sites [3]. This mechanism has not been replicated in C- allele carriers [10,21]. Similar results have been found in human hippocampal cell lines exposed to glucocorticoids during proliferation and differentiation [22]. In turn, intron 7 demethylation is higher in T-allele carriers [21], which reinforces the idea of allele-specific changes in methylation patterns [27–29].”
C: providing a brief comparison with other genes involved in the stress response (for example, NR3C1 or CRHR1) could give more context.
R: We appreciate the reviewer's comment. Briefly, we have included some recent evidence about the implications of NR3C1 and CRHR1 in PTSD, taking into account the most consistent findings in the disorder pathophysiology. “FKBP5 is a central determinant of HPA axis functioning, therefore, its functioning is of major relevance in stress-related disorders such as acute stress disorder and PTSD. Likewise, a number of other genes, such as CRHR1 and NR3C1, which encode for CRHR receptor and GR, respectively, are essential as well. In PTSD there is an increase in CRH secretion associated with an increase in negative feedback to the HPA axis [53]. This phenomenon will also be determined by the presence of genetic risk variants and their epigenetic changes for the development of HPA axis dysfunction [54].” (new references: Daskalakis et al. 2013 and Alahmad et al. 2025).
C: a schematic representation of the "double hit theory" would be very helpful for visualisation.
R: We appreciate the reviewer's comment. We have included “Figure 1” depicting the double hit theory.

Round 2
Reviewer 3 Report
Comments and Suggestions for Authors
The revised manuscript may be considered for publication.
Comments on the Quality of English LanguageI am not qualified to assess the quality of the English language in the manuscript, but the presented text did not pose any issues regarding comprehension